# Dissection of Common Rust Resistance in Tropical Maize Multiparent Population through GWAS and Linkage Studies

**DOI:** 10.3390/plants13101410

**Published:** 2024-05-18

**Authors:** Linzhuo Li, Fuyan Jiang, Yaqi Bi, Xingfu Yin, Yudong Zhang, Shaoxiong Li, Xingjie Zhang, Meichen Liu, Jinfeng Li, Ranjan K. Shaw, Babar Ijaz, Xingming Fan

**Affiliations:** 1Institute of Food Crops, Yunnan Academy of Agricultural Sciences, Kunming 650205, China; jiangfuyansxx@126.com (F.J.); biyq122627@163.com (Y.B.); xingfuyin626@163.com (X.Y.); mikezhangy@yahoo.com (Y.Z.); ranjanshaw@gmail.com (R.K.S.); babarijazpbg@gmail.com (B.I.); 2Institute of Resource Plants, Yunnan University, Kunming 650500, China; lilinzhuo0606@163.com (L.L.); 15987701739@163.com (S.L.); xingjiezhang2022@163.com (X.Z.); shirleyliu1028@163.com (M.L.); jinfengli1020@163.com (J.L.)

**Keywords:** tropical maize, multiparent populations common rust, QTL, GWAS, candidate genes

## Abstract

Common rust (CR), caused by *Puccina sorghi*, is a major foliar disease in maize that leads to quality deterioration and yield losses. To dissect the genetic architecture of CR resistance in maize, this study utilized the susceptible temperate inbred line Ye107 as the male parent crossed with three resistant tropical maize inbred lines (CML312, D39, and Y32) to generate 627 F_7_ recombinant inbred lines (RILs), with the aim of identifying maize disease-resistant loci and candidate genes for common rust. Phenotypic data showed good segregation between resistance and susceptibility, with varying degrees of resistance observed across different subpopulations. Significant genotype effects and genotype × environment interactions were observed, with heritability ranging from 85.7% to 92.2%. Linkage and genome-wide association analyses across the three environments identified 20 QTLs and 62 significant SNPs. Among these, seven major QTLs explained 66% of the phenotypic variance. Comparison with six SNPs repeatedly identified across different environments revealed overlap between *qRUST3-3* and *Snp-203,116,453*, and *Snp-204,202,469*. Haplotype analysis indicated two different haplotypes for CR resistance for both the SNPs. Based on LD decay plots, three co-located candidate genes, *Zm00001d043536*, *Zm00001d043566*, and *Zm00001d043569*, were identified within 20 kb upstream and downstream of these two SNPs. *Zm00001d043536* regulates hormone regulation, *Zm00001d043566* controls stomatal opening and closure, related to trichome, and *Zm00001d043569* is associated with plant disease immune responses. Additionally, we performed candidate gene screening for five additional SNPs that were repeatedly detected across different environments, resulting in the identification of five candidate genes. These findings contribute to the development of genetic resources for common rust resistance in maize breeding programs.

## 1. Introduction

Maize common rust is caused by the maize stalk rust *Puccinia sorghi* Schw during maize growth and development and is widely distributed in tropical, subtropical, and temperate areas. It develops easily at 15–25 °C and 98% humidity; it reduces photosynthesis in the leaf area and foliar failure by producing spots on the leaves, resulting in incomplete filling of the kernels and lower yields. Losses due to common rust in maize have been reported to range from 12 to 75% [1,2,3,4], and, due to ecological and high economic losses, breeding resistant plants is the best way to combat the disease and improve yields [5,6].

The breeding of disease-resistant varieties begins with the identification of resistance loci. Previous studies have shown that maize has qualitative and quantitative resistance to common rust [6,7,8]. Early studies were conducted to improve maize resistance by identifying the dominant resistance (Rp) gene; however, because dominant genes do not possess horizontal resistance, the loss of maize resistance is often accompanied by mutations in a specific *P. sorghi* race [5,9]. Consequently, the focus of research on common rust resistance in maize has shifted to non-specific quantitative resistance.

Multiple studies have successfully identified quantitative trait loci (QTL) for resistance to common rust in temperate maize germplasms through linkage mapping, and QTLs have been reported across all 10 maize chromosomes. One study identified a QTL in the 2.05–2.06 bin interval in a European dent maize population, explaining 25.5% of the phenotypic variance consistently found across different genetic backgrounds. Another study found a QTL in the 7.01 bin interval, explaining 23.6% of the phenotypic variance, which was also observed in different genetic combinations. However, another study discovered a QTL in the 3.04 bin interval (98 mb), explaining 20% of the phenotypic variance, which overlapped with previous research [7,10,11]. These findings suggest the possibility of potential existence of overlapping QTL regions governing the resistance to common rust in maize across diverse genetic backgrounds. Enhanced resistance to common rust in maize occurs when multiple partial resistance QTLs are combined. Accumulating disease-resistant QTLs can enhance plant resistance, underscoring the importance of identifying novel QTLs to further enhance maize resistance to common rust [12,13,14,15,16,17,18].

Compared to linkage analysis, genome-wide association analysis (GWAS) offers a higher mapping resolution and is therefore widely utilized in plant molecular breeding studies. However, this method often generates false-positive associations. Therefore, to obtain accurate results, it is crucial to eliminate these false associations. Considering the population structure during GWAS is the most effective approach for reducing these false associations [19,20,21,22,23,24,25,26]. For instance, GWAS analysis was conducted for resistance to common rust in a population of 274 temperate maize inbred lines. Four SNPs were identified, located on chromosomes 2 (59,014,463 bp), 3 (21,262,214 bp and 56,476,524 bp), and 8 (107,796,411 bp). Subsequently, four candidate genes (*GRMZM2G437912*, *GRMZM2G031004*, *GRMZM2G409309*, and *GRMZM2G089308*) were selected based on their association with these SNPs [27]. When GWAS and linkage analysis are combined, the resulting SNP loci fall within QTL regions. These outcomes are often more reliable than those obtained using GWAS or linkage analysis alone. Based on current research findings, the simultaneous application of these two methods can effectively elucidate the genetic mechanisms of quantitative traits [28,29].

In recent years, tropical maize germplasms have been used in studies of common rust resistance loci due to their broad genetic base [5,9,12]. However, the utilization of a multiparent population derived from common temperate and different tropical parents in advance generation (F7) for linkage mapping and genome-wide association analysis (GWAS) for common rust resistance in maize has not been previously attempted. Crosses between temperate and tropical lines can maximize genetic variations, provide clear genetic backgrounds for each plant, enabling better tracking and understanding of the genetic mechanisms underlying disease susceptibility and resistance. The relative stability of the F_7_ generations can narrow the QTL range, thereby enhancing the accuracy. Previous GWAS studies on common rust resistance in maize have predominantly utilized natural populations, including 296 tropical inbred lines, and 282 diverse inbred lines, as well as 380 tropical and subtropical inbred lines. These studies have all used GWAS models to account for population structure effects but have not explored the use of constructed populations to mitigate these effects. While natural populations offer time and cost benefits and directly reflect real ecological conditions, issues such as population structure effects, lack of genetic variation control, and environmental noise can compromise the GWAS accuracy. The present study addressed these challenges through multiparent population construction, ensuring more precise results [30,31,32,33,34].

In this study, three tropical inbred parents (CML312, D39, and Y32) that showed resistance to maize common rust were crossed with a common temperate susceptible backbone inbred parent, Ye107. A multiparent population comprising three F_7_ RIL subpopulations was constructed from these crosses. These three subpopulations were phenotyped in multiple environments to assess their response to common rust, and they were genotyped using the genotyping-by-sequencing (GBS) approach. The main objectives of this study were to identify QTLs and SNP loci that are significantly associated with common rust in different environments and to screen candidate genes associated with common rust.

## 2. Results

### 2.1. Phenotyping of Common Rust Resistance in RILs

Three RIL subpopulations were investigated for their responses to common rust in three different environments, and phenotypic data were recorded. Descriptive statistics data showed that the co-efficient of variation of the common rust disease in the three RIL subpopulations in the three environments ranged from 29.4% to 49.7%, with a high degree of inter-sample variability (Table 1). The kurtosis and skewness of the common rust phenotype ranged from −1 to 1 in absolute values across environments, indicating a small degree of deviation, and the data were normally distributed. The genotypic variance of the three subpopulations and genotypic variation due to environmental interactions were statistically significant (*p* < 0.05). The heritability of common rust disease was high in all three populations (85.7–92.2%), indicating that the trait is more influenced by genotype and less by environment.

Figure 1A illustrates the phenotypic differences among the three subpopulations across three distinct environments. Overall, significant differences were observed among subpopulations, whereas differences within the same population across different environments were less pronounced. Pearson correlation analysis with a significance level of *p* < 0.001 revealed strong correlations (0.773–0.856) between the overall performance of RILs within the same population across different environments. (Figure 1B). The strong correlation indicated that the RILs responded consistently to common rust infection in different environments, which not only indicates the high heritability of the common rust disease reactions but also reflects the high reliability of this experiment, providing a solid foundation for subsequent QTL/SNP mapping related to common rust resistance in maize.

### 2.2. QTL Mapping of Common Rust Resistance in Three RIL Subpopulations

To identify QTLs for common rust resistance, we performed QTL mapping in three RIL subpopulations (Pop1, Pop2, and Pop3) in different environments. SNP markers with ≥10% missing rates and loci with minor allele frequencies below 5% were excluded from the analysis. QTL analysis was conducted separately for these three subpopulations, identifying a total of 20 QTLs located on chromosomes 1, 2, 3, 4, 6, and 8 (Appendix A). Due to the inclusion of QTLs identified in different environments and the elimination of environmental effects, the best linear unbiased prediction (BLUP) information of the three subpopulations was used for QTL mapping, as depicted in Figure 2 and Table 2 for clarity and ease of reference.

Seven QTLs were distributed on chromosomes 2, 3, 4, and 6, and the LOD values ranged from 3.1 to 6.63, with phenotypic variance (R^2^) ranging from 8% to 12% (Table 2). Among these QTLs, the LOD value of *qRUST3-3* was the highest, at 5.39, and the phenotypic variation was 11%. Therefore, we concluded that this QTL has the potential to become the main QTL for maize resistance to common rust. *qRUST6-1* explained the highest phenotypic variance of 12%, and the additive effects of *qRUST2-1*, *qRUST3-1*, and *qRUST3-2* were negative, indicating that these three QTLs could negatively affect maize resistance to common rust.

### 2.3. SNP Characterization, Phglogenetic Tree, Principal Componenet Analysis Population and Heat Map Construcction

A heat map illustrating the marker density across the ten maize chromosomes is shown in Figure 3. The numbers of SNPs on chromosomes 1 to 10 were 1,223,552; 65,803; 63,745; 73,660; 56,253; 46,108; 52,458; 48,899; 43,833; and 42,070, respectively. Chromosome 1 had the highest number of SNP markers and chromosome 10 had the lowest number (Figure 3A). In the filtered SNP dataset, the average SNP missing rate was 0.2, the average minor allele frequency (MAF) was 0.5, and the filtered SNPs were used for subsequent genome-wide association studies (Figure 3B,C). The raw SNP dataset for each RIL population was used for the linkage disequilibrium (LD) decay analysis. We calculated the LD decay for all the populations and found, at a physical distance of 20 kb at the r2 threshold, a value of 0.2 (Figure 3D). Thus, candidate genes were screened 20 kb upstream and downstream of the significantly associated SNPs.

The phylogenetic tree showed that all 627 RILs could be categorized into three clusters (Figure 4). Overall, the phylogenetic tree, principal component analysis, and correlation heat map revealed that kinship among the RILs was consistent, and the population was divided into three clusters. The numbers of lines in the three clusters were 180, 223, and 224. The results of the first two principal component analyses validated the three clusters (Pop1, Pop2, and Pop3) identified by the phylogenetic tree. The small overlap in the center of the principal component analysis plot was due to the presence of the common parent Ye107 in all three populations.

### 2.4. Genome-Wide Association Analysis of Three RIL Subpopulations

We employed 573,112 SNPs in four different environments for GWAS analysis. SNPs with minor allele frequency (MAF) ≥ 5% and r^2^ < 0.2 were used to identify significant SNPs. A total of 62 SNPs associated with common rust resistance were identified at a threshold value of −log_10_(p) > 4.5 (Figure 5; Appendix A). The 62 SNPs were distributed on chromosomes 2, 3, 4, 5, 6, 7, 8, and 10. Seventeen significant SNPs were identified in the BLUP environment, 17 in the 21JH environment, six in the 21YS environment, and 22 in the 22YS environment. Although only one SNP was distributed on chromosome 8, chromosome 3 had a higher distribution across all environments. 

Six SNPs were consistently identified across all the three environments. These SNPs, along with the ten candidate genes identified, are presented in Table 3. SNPs significantly associated with common rust resistance were detected on chromosomes 3 and 5 in the 21JH, BLUP, and 22YS environments, with the highest *p*-values for *Snp-224,639,688* on chromosome 3 in all three environments exceeding 5.7. On the other hand, chromosomes 8 and 10 detected SNPs associated with common rust resistance in the BLUP, 21JH, and 21YS environments. We screened candidate genes 20kb upstream and downstream of tSNPs based on linkage disequilibrium (LD) decay analysis. Among the 10 candidate genes screened, *Zm00001d043536*, *Zm00001d043566*, *Zm00001d043569*, *Zm00001d044303*, and *Zm00001d015778* were functionally annotated, whereas the remaining five candidate genes have not yet been annotated.

### 2.5. Analysis of Consistent Loci Identified by GWAS and QTL Mapping

To determine whether the two different approaches could jointly screen for candidate genes associated with common rust resistance, we compared the QTL mapping results with the GWAS results. As shown in Table 4, *Snp-203,116,453* and *Snp-204,202,469* on chromosome 3, identified through GWAS, were located within the interval of *qRUST3-3*. Therefore, we designated these two SNPs as the focus of our subsequent research. A search within 20kb upstream and downstream of the two significant loci revealed three co-located candidate genes (*Zm00001d043536*, *Zm00001d043566* and *Zm00001d043569*) associated with common rust resistance.

We further analyzed *Snp-203,116,453* and the candidate gene *Zm00001d043536* (Figure 6). The relative positions of this SNP and the candidate gene and the positions of the significant SNPs identified by GWAS on chromosome 3 are presented in Figure 6A,B. This locus has two haplotypes, CC and TT, and the plants with the CC haplotype showed better resistance to common rust in maize (Figure 6C). *Zm00001d043536* was identified as the first gene governing resistance to common rust in maize, which might be attributed to the present of abundant variation in the tropical parents used in this study. We compared the exon base sequences within the gene interval and found that exon 2 of the gene had three subversions common to tropical inbred lines, resulting in the transcription of two amino acids that were different from those of the temperate parent (Figure 6D). During RNA-seq, the gene was found to be expressed during the growth of both young and mature leaves, demonstrating its association with common rust resistance in maize (Figure 6E) [35].

Similarly, we analyzed another significant locus, *Snp-204,202,469*, and its associated candidate genes, *Zm00001d043566* and *Zm00001d043569*, and the results are shown in Figure 7. Figure 7A,B show the locations of the significant SNP and their associated candidate genes. We performed haplotype analysis of the SNP and found that there were two haplotypes, GG and AA, and the plants of the GG haplotype showed better resistance to common rust of maize (Figure 7C). Our study of the four parents revealed widespread base substitutions and changes in the amino acids in two candidate genes that differed between the tropical and temperate parents (Figure 7D,F). The candidate gene *Zm00001d043566* showed the highest expression in the leaves of the plant (Figure 7E), and the candidate gene *Zm00001d043569* showed increased expression in the leaves and internodes after pollination (Figure 7G), which could prove that the two candidate genes are related to resistance to common rust in maize.

## 3. Discussion

In this study, three tropical inbred lines were used as female parents, and a temperate inbred line was used as the common male parent to construct three F_7_ subpopulations, totaling 627 RILs. Phenotypic analysis revealed a spectrum of reactions ranging from high susceptibility to high resistance across all populations, indicating the existence of polygenic resistance in the selected materials. The significant genotype-environment interaction variance in the three subpopulations underscores its importance in maize resistance to common rust. The high heritability observed across populations can be attributed to the abundant genetic variation resulting from the hybridization of temperate and tropical germplasms. 

### 3.1. The Comparison of the Results of This Study with Those of Previous Studies

In recent years, studies aiming to identify resistance loci against common rust disease in tropical maize have discovered new resistance loci and candidate genes owing to the rich genetic variation present in tropical maize breeding populations. However, these studies have primarily focused on analyzing natural or early generation populations, and investigations into resistance to common rust disease in tropical maize using advanced generation populations are lacking. This study addressed this gap by employing a different approach through population construction, aiming to enhance the accuracy of linkage analysis and genome-wide association analysis results [5,9,12]. The construction advanced generation populations offer several advantages over natural populations: (1) Although natural populations effectively reflect real biodiversity and genetic backgrounds, their complex population structure can compromise the reliability of GWAS analysis. This issue can be addressed through the population design itself. During the formation of recombinant parents, the mixing of parental alleles can mitigate the population structure within each population, thereby reducing the occurrence of false-positive associations and enhancing GWAS resolution. (2) Environmental conditions experienced by individuals in natural populations are difficult to replicate. However, high-generation population construction allows for better environmental control by conducting multiple replicate experiments to improve phenotype accuracy and subsequently enhance GWAS resolution.

Using a temperate inbred line as a common parent and hybridization with tropical inbred lines resulted in a broader genetic base than that of the tropical natural population. Additionally, constructing advanced generation populations may require more time compared to early generation populations, as they undergo more rounds of selfing, resulting in more fixed genotypes and narrower QTL intervals. These studies can lead to the identification of genetic loci and selection of candidate genes, laying the foundation for subsequent gene function validation. Therefore, enhancing the reliability of GWAS analysis and narrowing QTL intervals are crucial for improving the precision of mapping. The present study identified overlapping QTLs and SNPs reported in previous studies, along with novel significant loci that were not previously identified. To enable a clear comparison of these findings, they are succinctly summarized in Table 5.

Utilizing different populations to identify QTLs within the same genomic interval demonstrated the potential of these QTLs as major-effect QTLs. Previous studies have conducted QTL mapping using five F_3_ early generation populations, wherein *qCR3-113* overlapped with *qRUST3-3* in this study, as detailed in Table 6. A previous study used early generation populations (F_3_), resulting in a considerably large interval for *qCR3-113* (111.4 Mb). In contrast, *qRUST3-3* spanned a length of 34.7 Mb compared to the previous study, and the overlapping QTL in this study exhibited a higher LOD value (5.39) than the previous study (2.85) [5]. For studies aimed at identifying candidate genes, narrower intervals and increased LOD values represent a more accurate QTL, underscoring the advantage of the F_7_ population. In another study, 296 tropical maize inbred lines were used to identify QTLs distributed on chromosomes 1, 3, 5, 6, 8, and 10, among which the QTL on chromosome 6 (*Rp 6.1*) was found to be close to *qRUST6-1* identified in our study, with a distance of 564,094 bp [9].

This study considered eliminating population structure and environmental noise to enhance GWAS resolution during population construction to achieve accurate SNP identification. The criteria for SNP selection were based on repeatedly screening in different environments (Table 3). Haplotype analysis was conducted to determine whether the two co-located SNPs (*Snp-203,116,453* and *Snp-204,202,469*) were associated with resistance to common rust disease in maize, indicating that both loci play a significant role for the target trait (Figure 6C and Figure 7C). Additionally, *Snp-224,639,688* was 71,788 bp away from the *qCR3-113* interval, and *Snp-118,608,571* on chromosome 5 overlapped with the QTL *qCR5-51* STICHEL interval. Although Kibe et al. also identified QTLs associated with resistance to common rust disease on chromosomes 8 and 10, the SNPs discovered in this study did not overlap with these regions [5]. The candidate gene *GRMZM2G060540*, which was identified through the screening of *S3_147013779* in the study by Kibe et al., is of an uncharacterized nature, whereas our study suggests that three candidate genes (*Zm00001d043536*, *Zm00001d043566*, and *Zm00001d043569*) identified through co-located SNPs could provide a new direction of research on this stable QTL for common rust resistance in maize.

### 3.2. Functional Analysis of Candidate Genes Associated with Common Rust Resistance

One of the three candidate genes, *Zm00001d043536*, encodes heat stress transcription factor c1-b of the HSF family. HSF transcription factors regulate the expression of abscisic acid (ABA) [36], jasmonic acid (JA) [37], indole-3-acetic acid (IAA), and other plant hormones [38] which mediate gene activation under heat or other stress conditions to enhance plant stress tolerance. Thirty HSF proteins have previously been identified in maize. *Zm00001d043566* is a member of the STICHEL-3 protein family. The (STI) gene encodes a protein containing a domain with sequence similarity to the ATP-binding portion of the γ subunit of DNA polymerase III of true bacteria [39], which has been shown to be associated with the regulation of trichome branching number in *Arabidopsis* [40]. In maize, this function of this gene has been correlated with the number of trichome branches through gene homology studies with *Arabidopsis* [41]. The *Zm00001d043569* gene encodes the WRKY29 transcription factor, which has been shown to regulate ethylene biosynthesis and response in *Arabidopsis thaliana* [42]. Previous research has indicated that excessive immune responses in plants can adversely affect plant growth and development. Plant-induced ethylene synthesis acts as a negative regulator of immune responses, alleviating their impact on plants [42]. Furthermore, in GWAS, two annotated genes, *Zm00001d044303* and *Zm00001d015778*, were identified near the significant SNPs. *Zm00001d044303*, located near *Snp-224,639,688* on chromosome 3, encodes a myosin protein crucial for cytokinesis and intracellular movement. *Zm00001d015778*, near *Snp-118,608,571* on chromosome 5, encodes leucine repeats associated with plant innate immunity.

### 3.3. The Application of Parental Lines Used in the Present Study in Commerial Breeding

Tropical maize germplasms have long been recognized for their wide range of disease resistance and have been extensively utilized in the breeding of disease-resistant maize varieties. Historically, maize breeders have introgressed disease resistance genes from these tropical inbred lines to improve resistance in temperate germplasms. Ye107, a backbone inbred line of temperate origin, plays a crucial role in the production of key corn varieties, such as Yunrui8. CML312, selected from CIMMTY, is a high-quality tropical inbred line that has contributed to the development of disease-resistant hybrids, such as ‘Yunrui2’. D39 is an excellent tropical inbred line that produced the hybrid D‘edan5’, which was inoculated and identified as highly resistant to rust (disease class 1) by the College of Plant Protection of Anhui Agricultural University (AAU). Y32, a high-quality inbred line derived from the classic tropical germplasm Suwan, has been instrumental in breeding a high-quality and stress-resistant hybrid, ‘Yunrui1’. However, as common rust resistance is a quantitative trait controlled by multiple minor genes, it is challenging for maize breeders to introgress minor resistance genes from tropical germplasms into the targeted maize germplasm. Many of these minor genes provide partial resistance and may not be sufficient to confer durable resistance, necessitating introgression of multiple minor resistance loci. Nevertheless, advances in molecular marker techniques have refined the process of selection for disease resistance. Hence, additional candidate loci associated with common rust resistance can be identified. The present study has laid a strong foundation for the development of common rust-resistant inbred lines and hybrids in maize. 

## 4. Conclusions

In this study, we used three F_7_ RIL subpopulations derived from crosses between three tropical inbred lines and a common temperate parent, comprising 627 recombinant inbred lines, to elucidate the genetic architecture of maize resistance to common rust. Linkage mapping identified seven major QTLs that explained 66% of the phenotypic variance. Using a genome-wide association study (GWAS), we identified SNPs consistently observed across six distinct environments and compared them with QTL mapping results. Two SNPs, *Snp-203,116,453* and *Snp-204,202,469*, on chromosome 3 were found to fall within the interval of QTL *qRUST3*-3. Based on the positions of these two SNPs and the linkage disequilibrium decay plots, we identified three candidate genes (*Zm00001d043536*, *Zm00001d043566*, and *Zm00001d043569*). A comparison with previous studies revealed an overlap of *qRUST3-3* identified in the present study, albeit with significantly reduced intervals, and the identification of three new candidate genes with protein functional annotations. Our findings contribute to elucidating the mechanisms underlying common rust resistance and lay the groundwork for future functional validation studies.

## 5. Materials and Methods

### 5.1. Experimental Materials and Field Experiment Design

In this study, the tropical maize inbred lines Y32, CML312, and D39 were used as female parents, the temperate inbred line Ye107 was used as the common male parent for hybridization, and all three inbred lines showed resistance to common rust. The F_1_ plants were self-pollinated for six generations through the single seed descent method, and three RIL subpopulations were developed: Pop1 (CML312×Ye107), Pop2 (D39×Ye102) and Pop3 (Y32×Ye107) (Table 7). The three populations pop1, pop2, and pop3 consisted of 180, 223, and 224 RILs, respectively, totaling 627 RILs. In Yunnan Province, China, the common rust in maize typically breaks out in July each year. During this period, the selected locations, Jinghong City (latitude 21.850° N, longitude 100.931° E, elevation 539 m) and Yanshan (latitude 23.485° N, longitude 104.100° E, elevation 1572 m), experienced temperatures around 25 °C and high relative humidity exceeding 81%, creating favorable conditions for the growth and spread of common rust spores. Subsequently, phenotypic evaluations of the common rust disease were conducted at these locations. All RILs were planted in 2021 in Jinghong City (21JH), Yunnan Province, and Yanshan County (21YS), Wenshan Prefecture, Yunnan Province. In 2022, the experiment was replicated in YS County, Wenshan Prefecture, Yunnan Province (22YS). A randomized complete block design was used for all experiments. Each experimental plot was composed of a 4-m-long row, with a row spacing of 0.70 m and a plant spacing of 25 cm [43] The trials were conducted according to standard agronomic practices (Figure 8).

### 5.2. Common Rust Disease Evaluation

Three populations were evaluated for common rust response under sustained high natural disease pressure starting at week 2 after maize dispersal, and maize common rust incidence levels were surveyed at 7-day intervals three times. Resistance scores were determined for each RIL based on the percentage of infected area to total area, and the scores were based on specific criteria listed in Table 8 [43].

### 5.3. Phenotypic Data Analysis

After preliminary processing of the phenotypic data of the three RIL subpopulations collected at two locations over two years, SPSS Statistics was used to analyze the data. First, descriptive statistical analyses, including mean, standard deviation, variance, skewness, kurtosis, and coefficient of variation, were calculated. Using SPSS Statistics. Kurtosis and skewness estimates confirm a normal distribution for common rust disease reaction. Broad-spectrum heritability was calculated using the following method [44,45]:$$H^{2}\;=\;\frac{\sigma_{g}^{2}}{\sigma_{g}^{2}+\frac{\sigma ge^{2}}{e}+\frac{\sigma \varepsilon^{2}}{re}}\times 100\%$$
where $\sigma_{g}^{2}$ is the genetic variance, *σge*^2^ is the variance due to environment × genotype interactions, *σε*^2^ is the residual, *e* represents the number of environments or locations, and *r* represents the number of replications per location. *H*^2^ identifies the degree of variation in a phenotypic trait; the more significant the *H*^2^, the more the trait is influenced by the genotype and the less it is influenced by the environment.

Estimation of breeding values: BLUP values for each trait in all environments were obtained for each inbred line using a linear mixed model in R (v.3.6.1) (http://www.r-project.org/) with the lme4 package. The formula used for calculating the BLUP values is as follows [45]:$$Y_{ijk}=\mu +G_{k}+E_{i}+R_{j(i)}+{EG}_{ik}+\varepsilon_{ijk}$$
where *Y_ijk_* is the observed value of the *j*th repetition of the *k*th genotype in the *i*th environment, *µ* is the overall mean, *G_k_* is the effect of the kth genotype, *E_i_* is the effect of the *i*th environment, *R*_*j*(*i*)_ is the effect of the *j*th repetition nested in the *i*th environment, *EG_ik_* is the effect of the interaction between the *i*th environment and *kth* genotype, and *ε_ijk_* is the effect of experimental error.

Violin and correlation heat maps were plotted in the hiplot website (https://hiplot.com.cn/home/index.html, accessed on 15 August 2023).

### 5.4. DNA Extraction and Genotyping-by-Sequencing (GBS)

Genomic DNA was extracted from young maize leaves of RILs of the three subpopulations using the cetyltrimethylammonium bromide (CTAB) method [45]. The DNA was digested using the restriction endonucleases *Mspl* and *Pstl* (New England Biolabs, Ipswich, MS, USA) and then ligated with a barcode adapter using T4 ligase (New England Biolabs). All ligated samples were mixed and purified using the QIAquick PCR Purification Kit (QIAGEN, Valencia, CA, USA). Primers complementary to the two adapters were used for PCR amplification. The PCR products were purified and quantified using the Qubit dsDNA HS Assay Kit (Life Technologies, Grand Island, NY, USA). PCR products between 200 and 300 bp in size were selected using the Egel system (Life Technologies, USA), and library concentrations were estimated using the Qubit 2.0 fluorometer and the Qubit dsDNA HS assay kit (Life Technologies, USA). GBS libraries were constructed, and sequencing was conducted according to the GBS protocol [46] Sequencing was performed using an Ion Proton sequencer (Life Technologies, software version 5.10.1) and P1v3 chip. Final reads were generated using TASSEL v5.0 (https://github.com/Euphrasiologist/GBS_V2_Tassel5, accessed on 23 July 2023) [47]. Prior to TASSEL analysis, 80 polyadenylates (poly (A) bases) were appended to 30 ends of all the sequencing reads [48]. Using the Genome Analysis Toolkit software (GATK-3.8), SNPs were identified by aligning to the maize reference genome, B73 (B73_V4, ftp://ftp.ensemblgenomes.org/pub/plants/release-37/fasta/zea_mays/DNA, accessed on 23 July 2023) [49]. A total of 573,112 high-quality SNPs were annotated using ANNOVAR software (v2013-05-20) [50]. After mapping the filtered raw reads with the maize reference genome B73 (RefGen_v4) to discover SNPs, the filtering parameter for SNP data was set to MAF ≥ 0.05 to identify the high-quality SNPs.

### 5.5. QTL Mapping

The three populations were phenotyped to determine the best results based on whether the SNPs and QTLs overlapped or not. Populations and phenotypes with the best results were carefully selected and the results were displayed. Pop2 (D39×Ye107) exhibited overlapping of SNPs and QTLs. The SNPs were then used to construct a genetic linkage map using JoinMap4 software [51]. Linkage groups were formed using an LOD threshold of ≥5.0. QTLs for common rust resistance were identified using the composite interval mapping (CIM) method using Windows QTL Mapper v2.0 [52]. The LOD threshold was set based on 1000 random permutation tests with a significance level of *p* ≤ 0.05 [53]. The results show that QTLs with LOD thresholds ≥3 were considered significant. The percentage of phenotypic variation (PVE) explained by a single QTL was calculated as the square of the partial correlation coefficient (R^2^).

### 5.6. Structure Analysis

We used Tassel 5.0 for phylogenetic tree analysis, R4.2.1, for principal component analysis and correlation heatmap plotting, whereas the principal component analysis was plotted with the scatterplot package, and the correlation heatmap was plotted using the GAPIT package.

PopLDdecay 3.40 software and perl scripts were used to assess linkage disequilibrium (LD) and determine the number of markers required for GWAS, detection efficiency, and accuracy. The LD decay figure was drawn using default parameters.

### 5.7. Haplotype Analysis

Haploview v4.2 software was used for haplotype analysis of SNP loci. Box line plots were drawn using Origin2022.

### 5.8. Genome Wide Association Study

After Illumina NovaSeq6000 sequencing, BAM files were processed, and then GWAS analysis was performed using the mixed linear model (MLM) implemented in GEMMA software (https://www.xzlab.org/software.html, accessed on 26 August 2023). Parameters were set to plink-indep-pairwise 5050.2, −log10(p) > 4.5 [54], and Manhattan and QQ plots were generated [55].

### 5.9. Identification and Functional Annotation of Candidate Genes

Maize GDB (https://www.maizegdb.org/gbrowse, accessed on 26 August 2023) and Maize Reference Genome B73 (RefGen_v4) were used to search for genes associated with common rust resistance in maize. The screened genes were considered as candidate genes involved in common rust resistance in maize and then the functions of the screened candidate genes on Maize GDB were annotated and compared using NCBI (https://www.ncbi.nlm.nih.gov/, accessed on 26 August 2023).

### 5.10. Candidate Gene Expression Analysis

We performed a query on a public database (http://ipf.sustech.edu.cn/pub/zmrna/, accessed on 5 October 2023) to retrieve information related to the expression of three co-localized candidate genes identified by association analysis. FPKM values for the candidate genes under common rust stress were obtained.

## Figures and Tables

**Figure 1 plants-13-01410-f001:**
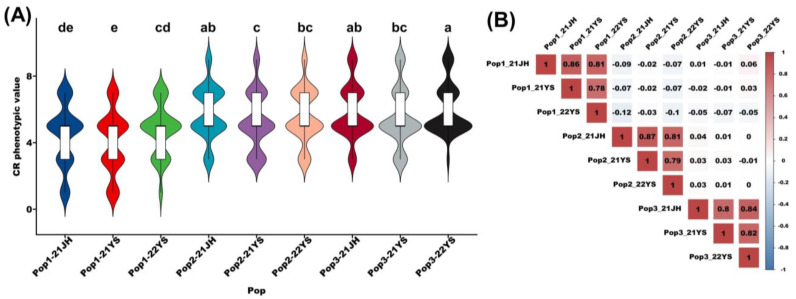
Distribution of rust traits in three subpopulations and their correlations. (**A**) Violin plots of the phenotypic distribution of the three populations. In (**A**), a–e shows the letter significance labelling method, in which the difference is not significant if there is a letter with the same labelling, and significant if there are letters with different labelling. (**B**) Heat map depicting the overall performance correlation of RILs within each of the three populations across different environments.

**Figure 2 plants-13-01410-f002:**
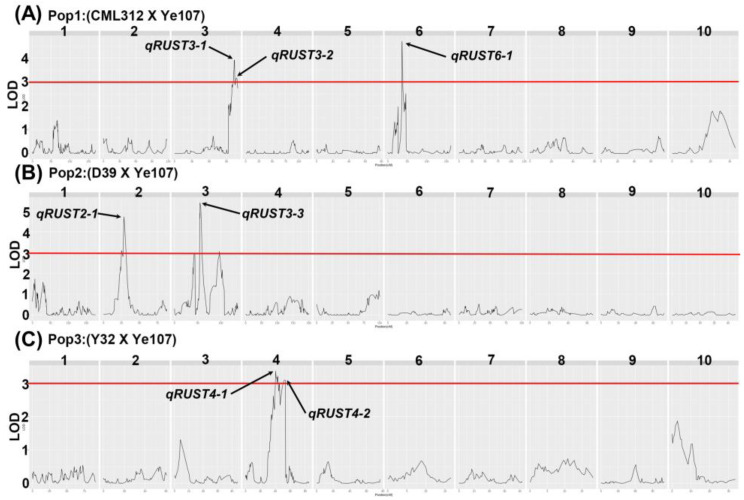
Logarithm-of-odds (LOD) profiles of BLUP value of the three subpopulations: (**A**) Log-of-odds (LOD) profiles of Pop1 (CML312×Ye107); (**B**) Log-of-odds (LOD) profiles of Pop2 (D39×Ye107); (**C**) Log-of-odds (LOD) profiles of Pop3 (Y32×Ye107). Numbers 1–10 above each figure indicate chromosomes.

**Figure 3 plants-13-01410-f003:**
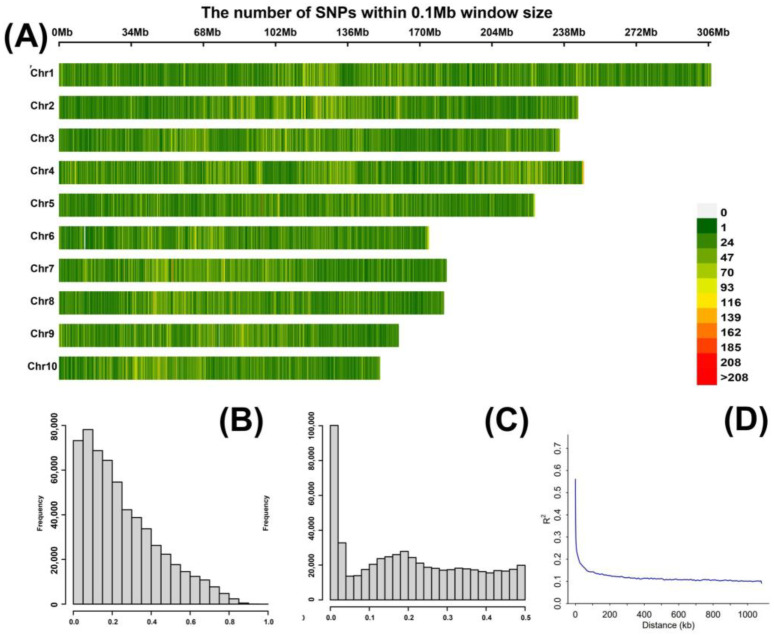
Phenotypic diversity in the atlas. (**A**) Density of chromosome-specific SNPs in the 0.1 Mb genomic interval. The number of SNPs is indicated on a green-to-red scale. (**B**) Distribution of minor allele frequencies of the SNPs. (**C**) Frequency distribution of missing genotypes. (**D**) Whole-genome LD in the entire panel based on 627 maize RILs.

**Figure 4 plants-13-01410-f004:**
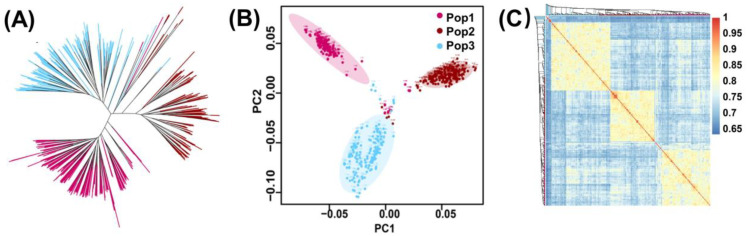
Genetic diversity analysis. (**A**) Phylogenetic tree of three populations. (**B**) Principal component analysis of 627 RILs. (**C**) Correlation heat map of 627 RILs.

**Figure 5 plants-13-01410-f005:**
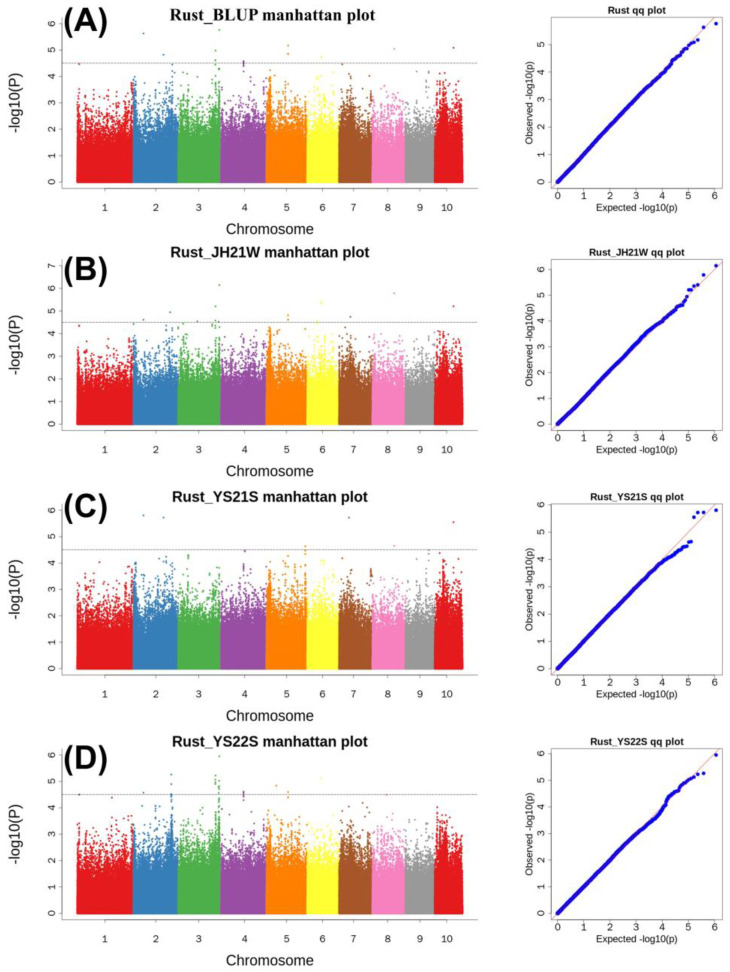
GWAS analyses in the three RIL populations. (**A**) Manhattan and Q-Q plots of the BLUP environment for common rust resistance; (**B**) Manhattan and Q-Q plots for the 21JH environment for common rust resistance; (**C**) Manhattan and Q-Q plots for the 21YS environment for common rust resistance; (**D**) Manhattan and Q-Q plots for the 22YS environment for common rust resistance.

**Figure 6 plants-13-01410-f006:**
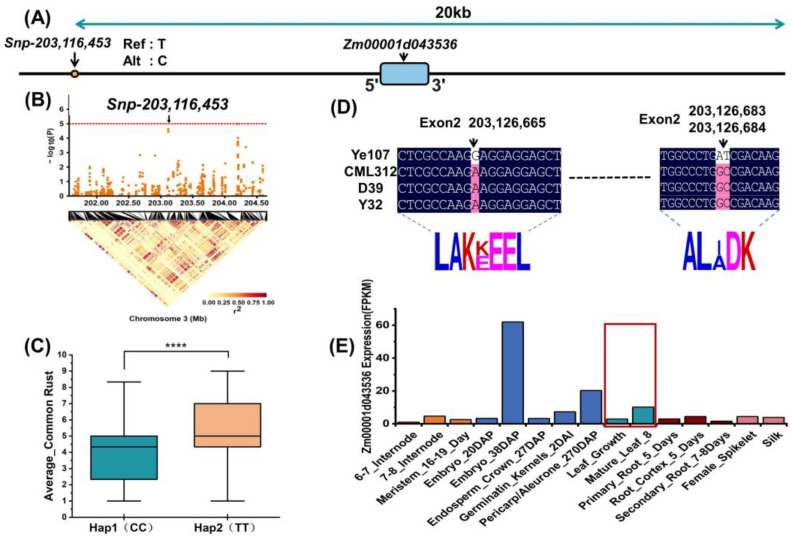
Identification of *Snp-203,116,453* candidate genes for common rust resistance. (**A**) Relative positions of SNP and candidate genes; (**B**) Positions of significant SNP in GWAS; (**C**) Differences between the two haplotypes in the overall resistance to common rust, with **** indicating *p* < 0.0001; (**D**) Base reversal causing amino acid changes in the candidate gene in different parental lines; (**E**) Expression levels of *Zm00001d043536* in various tissues. The red box highlights the expression of the gene in the leaves (DAP: days after pollination DAS: days after sowing.).

**Figure 7 plants-13-01410-f007:**
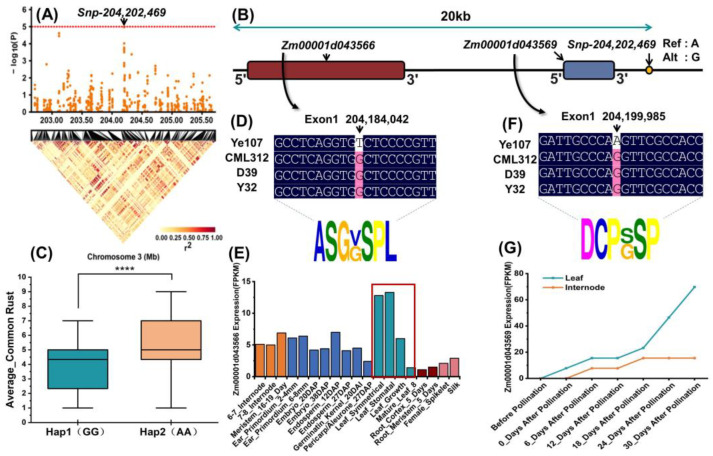
Identification of *Snp-204,202,469* and associated candidate genes for common rust resistance. (**A**) Position of significant SNPs in GWAS. (**B**) Relative position of SNP and candidate genes. (**C**) Difference between the two haplotypes in overall resistance to common rust, with **** indicating *p* < 0.0001. (**D**) Amino acid changes in candidate genes due to subversion in *Zm00001d043566*. (**E**) Expression levels of the *Zm00001d043566* gene in various tissues. The red box highlights the expression of the gene in the leaves. (**F**) Amino acid changes due to subversion in the candidate gene *Zm00001d043569*. (**G**) Expression of *Zm00001d043569* in leaves and internodes before and after pollination. (DAP: days after pollination DAS: days after sowing).

**Figure 8 plants-13-01410-f008:**
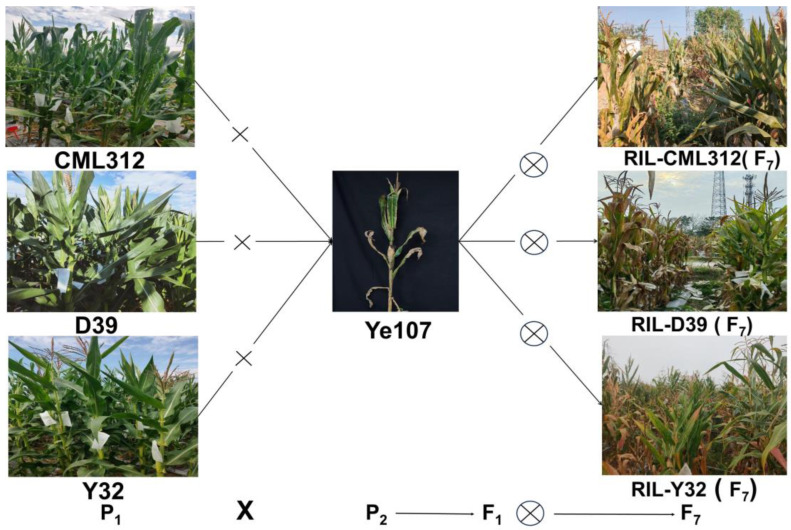
The population development is shown in the schematic diagram. CML312, D39, Y32 and Ye107 are the four parents, and Ye107 is the susceptible parent. The F_1_ generation produced from the cross of these four parents underwent six consecutive generations of self-crossing to obtain the F_7_ RILs.

**Table 1 plants-13-01410-t001:** Statistical analysis of common rust phenotype of three RILs populations.

Populations	Environments	Means	Standard Deviation	Skewness	Kurtosis	Coefficient of Variation (%)	Variance Components			Heritability (*H*^2^) (%)
							$\sigma_{g}^{2}$	$\sigma_{ge}^{2}$	$\sigma_{e}^{2}$	
Pop1	21JH	4.700	2.066	−0.330	−0.324	44.0	3.182 *	0.219 *	0.218	85.7
	21YS	4.322	2.147	0.218	−0.428	49.7				
	22YS	5.000	1.831	0.221	0.059	36.6				
Pop2	21JH	5.789	1.705	0.201	0.123	29.4	2.377 *	0.382 *	0.057	90.6
	21YS	5.439	1.871	0.248	−0.133	34.4				
	22YS	5.964	1.596	0.406	0.143	26.5				
Pop3	21JH	5.759	1.644	−0.121	−0.151	28.6	2.494 *	0.177 *	0.036	92.2
	21YS	5.268	1.817	0.166	−0.379	34.5				
	22YS	5.359	1.876	−0.332	−0.018	35.1				

* in the table indicates: *p* < 0.05.

**Table 2 plants-13-01410-t002:** QTLs identified for common rust resistance in maize in BLUP environment.

QTL	Chr	Position (cM)	Mapping Interval (cM)	LOD	Additive_Effect	R^2^ (%)
*qRUST2-1*	2	28.49	25.05–31.32	4.71	−0.48	0.09
*qRUST3-1*	3	103.72	101.71–103.72	3.92	−0.59	0.1
*qRUST3-2*	3	106.73	105.73–108.28	3.17	−0.54	0.08
*qRUST3-3*	3	54.96	54.41–59.02	5.39	0.7	0.11
*qRUST4-1*	4	40.43	40.12–43.27	3.37	0.46	0.08
*qRUST4-2*	4	52.39	51.39–53.39	3.1	0.45	0.08
*qRUST6-1*	6	36.39	36.39–38.39	4.71	0.92	0.12

**Table 3 plants-13-01410-t003:** Distribution of significant SNPs and candidate genes consistently identified by GWAS in different environments.

Environment	SNP	Chr	p-BLUP	p-21JH	p-21YS	p-22YS	Candidate Gene	Gene Annotation
BLUP 21JH 22YS	Snp-203,116,453	3	4.618	4.580	-	5.066	*Zm00001d043536*	Heat stress transcription factor C-1b
	Snp-204,202,469	3	4.978	5.208	-	5.223	*Zm00001d043566*	Protein STICHEL-like 3
							*Zm00001d043567*	-
							*Zm00001d043568*	-
							*Zm00001d043569*	WRKY-transcription factor 29
	Snp-224,639,688	3	5.763	6.145	-	5.949	*Zm00001d044303*	IQ_motif_EF-hand-BS
	Snp-118,608,571	5	5.169	4.812	-	4.596	*Zm00001d015778*	Leucine-rich repeat
BLUP 21JH 21YS	Snp-118,876,904	8	5.046	5.787	4.654	-	*Zm00001d010519*	-
	Snp-102,507,767	10	5.084	5.206	5.548	-	*Zm00001d025070*	-
							*Zm00001d025071*	-

**Table 4 plants-13-01410-t004:** Consistent loci detected in two different mapping approaches.

QTL/SNP	Chr	Position	Candidate Gene	Gene Annotation
*qRUST3-3*	3	172,823,884–210,543,887	*Zm00001d043536*	Heat stress transcription factorC-1b
Snp-203,116,453	3	203,116,453	*Zm00001d043566*	Protein STICHEL-like 3
Snp-204,202,469	3	204,202,469	*Zm00001d043569*	WRKY-transcription factor 29

**Table 5 plants-13-01410-t005:** Comparison of QTL and significant SNPs for common rust resistance in maize in this study with previous studies.

Chr	This Study		Previous Study		
	QTL/Snp	Position	QTL/Snp	Position	Reference
2	*qRUST2-1*	125,535,857–125,535,857	-	-	-
3	*qRUST3-1*	19,468,979–21,766,539	-	-	-
3	*qRUST3-2*	17,098,052–18,118,650	-	-	-
3	*qRUST3-3*	172,823,884–210,543,887	qCR3-113	113,425,715–224,567,900	[5]
5	*qRUST4-1*	121,288,117–128,564,645	-	-	-
5	*qRUST4-2*	94,866,787–94,866,787	-	-	-
6	*qRUST6-1*	99,941,104–110,962,870	-	-	-
3	Snp-203,116,453	203,116,453	qCR3-113	113,425,715–224,567,900	[5]
3	Snp-204,202,469	204,202,469	qCR3-113	113,425,715–224,567,900	[5]
3	Snp-224,639,688	224,639,688	-	-	-
5	Snp-118,608,571	118,608,571	qCR5-51	51,355,494–186,678,634	[5]
8	Snp-118,876,904	118,876,904	-	-	-
10	Snp-102,507,767	102,507,767	-	-	-

**Table 6 plants-13-01410-t006:** Comparison of chromosome 3 QTL and SNPs with previous studies.

**Chr**	**THIS STUDY**				**Distance (bp)**	**(Kibe et al., 2020)** [5]			
3	**Ye107 × D39(F_7_)**					**CZL0618 × LaPostaSeqC7-F71-1-2-1-1B(F_3_)**			
	**QTL/Snp**	**Pos**		**LOD**		**QTL/Snp**	**Pos**		**LOD**
	qRUST3-3	172,823,884 ~ 210,543,887	37.63Mb	5.39	-	qCR3-113	113,425,715 ~ 224,567,900	111.14Mb	2.85
	Snp-203,116,453	203,116,453		-	56,102,674	S3_147013779	147,013,779		-
	Snp-204,202,469	204,202,469		-	57,188,690				

**Table 7 plants-13-01410-t007:** Maize parental lines used in developing RIL subpopulations.

Parents	Pedigree	Ecological Type	Rust Resistance	Symptoms Scale of CR
Ye107	Derived from US hybrid DeKalb XL80	Temperate	Susceptible	9
CML312	S89500-F2-2-2-1-1-B*5-2-1-6-1	Tropical	Resistant	3
D39	Selected from Suwan1	Tropical	Highly Resistant	1
Y32	Suwan 1-SC9-S8-346-2(Kei 8902)-3-4-4-6	Tropical	Highly Resistant	1

CR = Common Rust.

**Table 8 plants-13-01410-t008:** Common rust disease scale used for screening the RILs of multiparent population.

Scale	Reaction Category	Symptoms
1	highly resistant	no or very few rust spots on the leaves, or lesion area less than 6% of the total leaf area
3	resistant	a small number of spots on the leaves, or lesion area comprising 6% to 25% of the total leaf area
5	moderately resistant	number of spots on leaves or lesion area covering 26% to 50% of the total leaf area
7	susceptible	number of spots on leaves or area of damage comprising 51% to 75% of the total leaf area
9	highly susceptible	large lesion area on leaves or 76% to 100% of leaf death

## Data Availability

The data presented in this study are available on request from the corresponding author.

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
