# Peer review of "Dissection of Common Rust Resistance in Tropical Maize Multiparent Population through GWAS and Linkage Studies"

_plants, 2024, doi:10.3390/plants13101410_

Round 1

Reviewer 1 Report

Comments and Suggestions for Authors

In general, the manuscript, Untitled Dissection of Common Rust Resistance in Tropical Maize Mul-2 Tiparent Populations through GWAS and Linkage Studies by Linzhuo Li et al., has interesting results and presents innovative tools combination that offers better analysis results to detect SNPs associated with

Common rust (CR) is a disease that considerably affects corn production.

However, I have some recommendations to improve the understanding of the research presented by the authors.

 Introduction

1.         Authors should expose what has already been done by other research groups and the benefits of those researchers in reducing or controlling the Common rust disease.

2.         The impact of the research presented concerning the previous ones.

Materials and methods

The authors do not clearly explain or describe the environmental conditions of the three populations in question or how they were obtained. It is a bit confusing how and why these environments were chosen, including the conditions of temperature, soil, humidity, etc. The nomenclature of each environment is not clear, which makes it difficult to read and understand the manuscript.

Results

Figure 1 is not cited in the manuscript, and it is not explained either. Also, each population's environment should be added in Figure 1 and Table 1 to clarify the results.

In Table 2, the authors mention the title BLUP environment, which has not been described in the manuscript before.

The results section contains too much information that can be repetitive. I suggest that the authors select the most relevant and leave the rest to the supplementary materials to better appreciate each figure.

The authors should clearly describe the section on SNPs and their relevance and do the same for the candidate genes in the following sections of the manuscript.

Discussion

In this section, the authors add tables of results that should be presented in the results section. The results section should be subdivided into headings and titles to understand the research work better. There are many figures and tables that could be reduced and make the manuscript easier to read.

Conclusions should be strong and clear, and the findings should be compared here with what is already available.

Authors should add the relevance of the SNPs because they mix the generated SNPs with candidate genes, which is the most important to fight against CR, and finally, what they are looking for.

In general, I suggest that the authors leave the most important sections, Results, Discussion, and Conclusion, to better appreciate the findings of this research.

Author Response

We genuinely appreciate being given the opportunity to revise our manuscript. The insightful comments provided by the reviewers have significantly improved both the quality and clarity of our research findings. We have carefully addressed each of the points raised by the reviewers.

Please check the uploaded word for the detailed modification response.

Reviewer 2 Report

Comments and Suggestions for Authors

The manuscript provides good insight into resistance to common rust in maize by combining phenotypic and marker data analyses. It is worth publishing after the authors look at my minor comments below.

Results

Table 1. Statistical analysis of Common Rust phenotype of three RILs populations.

Re-format the table to align Populations (Pop1, Pop2, Pop3), their environments and the variance components well. As it is now, the Pop and their corresponding environments is difficult to visualize.

L121-123: Are the authors referring to correlation between the populations means of the disease?. Please make this clearer by indicating it (“population mean”) in the L121-122.

Figure 1. “B Heat map of correlation between three groups in three environments”. Do you refer to the mean values of the three groups? Please state it in the footnote “Heat map of correlation between ….” 

L136: “…were identified in the three populations”. Is this for the three populations combined or separately? Make it clear.

QTL in pops

Could the varying number of population size between Pop1 (180)  versus Pop2 (223) and Pop3(224) influence the outcome of QTL and GWAS between the populations?

L139-142: Figure 2. I do not see any overlapping QTLs between the populations, but the populations performed “consistently” across the 3 environments (see L121-123). What could be the reason for this observation? Indicate it in the manuscript.

L146: “…the effect value of qRUST3-3…”. Do you mean the significance threshold (LOD) value (see table 2)?. The effect size/value is not the same as the significance threshold, LOD value. Check and update the statement “…effect value…”.

L155-156: Authors can use semicolons between two chr. position nrs. Eg., 1,223, 552; 65,803; 63,745; etc. to make it more readable.

Best regards

Author Response

(The authors gave the same response as above.)
